# Synthesis and Characterization of Chelating Hyperbranched Polyester Nanoparticles for Cd(II) Ion Removal from Water

**DOI:** 10.3390/molecules27123656

**Published:** 2022-06-07

**Authors:** Faten Alregeb, Fawwaz Khalili, Bassam Sweileh, Dalia Khalil Ali

**Affiliations:** 1Faculty of Science Arts and Sciences, Middle East University, Amman 11610, Jordan; fregab@meu.edu.jo; 2Department of Chemistry, Faculty of Science, The University of Jordan, Amman 1118, Jordan; fkhalili@ju.edu.jo (F.K.); sweilehb@ju.edu.jo (B.S.); 3Faculty of Allied Medical Sciences, Isra University, Amman 11622, Jordan

**Keywords:** hyperbranched polyester, chelating polyester, nanoparticles, interfacial polymerization, metal ion uptake, cadmium

## Abstract

Chelating hyperbranched polyester (CHPE) nanoparticles have become an attractive new material family for developing high-capacity nanoscale chelating agents with highly branched structures and many functional groups in the main chains and end groups that can be used to remove heavy metals from water. In this study, a hyperbranched polyester with a particle size of 180–643 nm was synthesized with A_2_+B_3_ interfacial polymerization, using dimethylmalonyl chloride as the difunctional monomer (A_2_) and 1,1,1-tris(4-hydroxyphenyl)ethane (THPE) as the trifunctional monomer (B_3_). FTIR and NMR were used to characterize the CHPE and confirm the structure. The CHPE nanoparticles were generally considered hydrophilic, with an observed swelling capacity of 160.70%. The thermal properties of the CHPE nanoparticles were studied by thermal gravimetric analysis (TGA) with 1% mass loss at temperatures above 185 °C. The XRD of the CHPE nanoparticles showed a semi-crystalline pattern, as evident from the presence of peaks at positions ~18° and 20°. The nature of the surface of the CHPE was examined using SEM. Batch equilibrium was used to investigate the removal properties of the CHPE nanoparticles towards Cd(II) ions as a function of temperature, contact time, and Cd(II) concentration. The Cd(II) ion thermodynamics, kinetics, and desorption data on the CHPE nanoparticles were also studied.

## 1. Introduction

Environmental pollution caused by toxic heavy metal ions is a growing problem worldwide because of the impact of heavy metals on the environment and human health. For example, cadmium is a highly toxic industrial and environmental pollutant, mainly discharged into the environment from several industrial activities. The human health effects caused by Cd are human carcinogen emphysema, liver, hypertension, testicular atrophy, and renal damage [1,2].

Many techniques for removing heavy metal ions from aqueous solutions have already been developed to address this concern, including precipitation, ion exchange, electrochemical treatments, coagulation, membrane filtration, and adsorption. Among these strategies, the adsorption method is one of the most effective methods for removing heavy metal ions from aqueous systems. Different nanomaterials have been applied as adsorbents, including activated carbons, graphene materials, metal oxides, clays, and polymers [3,4].

Chelating polymers have received much attention because of their intrinsic environmentally harmless and degradable properties. Various functional groups can be used as chelators, such as phosphorous acids, carboxylates, oximes, and 1,3-diketones [5,6,7]. Chelating polymers provide several applications in a wide range of fields [8], including the preconcentration of trace elements from aqueous solutions, selective metal ion determination [7,9,10,11,12], antidotes for some types of metal ion poisoning, and water treatment for the removal of heavy metal ions [13].

HPs are three-dimensional dendritic macromolecules with an extremely branching topology. HPs have received considerable attention because of their distinct physical and chemical properties. HPs often present superior physical, chemical, and mechanical properties compared to linear polymers. They have been applied in biomedical materials, analytical and detection chemistry, polymer coatings, processing aids, optoelectronic materials, polymer-derived ceramic devices, and treatments for the removal of metals from water [14,15,16,17,18,19,20].

Chelating hyperbranched polyester nano-surfaces have become attractive in research due to their highly branched structures and many functional groups in the main chains and end groups. Therefore, this new family of chelating polymers with three-dimensional molecular architecture, including dendrimers and hyperbranching, is unique for its ability to develop high-capacity nanoscale chelating agents [18,21,22,23].

Polymeric nanoparticles are typically produced using one of two methods. The first one is the preparation of nanoparticles from the dispersion of the preformed polymer, and the second is the preparation of nanoparticles from the direct polymerization of monomers [24,25,26,27].

Interfacial polymerization is a type of step-growth polycondensation that occurs at the interface of two immiscible solvents. This method can directly synthesize polymeric nanosized particles ranging from 100 to 900 nm. In this technique, polymer formation occurs at or near the liquid–liquid interface when two solutions are mixed [28,29,30,31]. 

This work aims to synthesize and characterize novel chelating hyperbranched polyester nanoparticles prepared by interfacial polycondensation. Polyester is expected to have important properties for the removal of metal ions from water because it contains the malonate chelating group. Therefore, its uptake towards Cd(II) ions was studied in this work as a function of variables such as temperature, contact time, and concentration. In addition, the thermodynamics, kinetics, and desorption data of the metal ions on the polyester were also studied. 

## 2. Results and Discussion

### 2.1. Synthesis of Chelating Hyperbranched Polyester Based on THPE by Interfacial Polycondensation

Polyester was prepared by an interfacial polymerization reaction between dimethylmalonyl chloride and THPE, as shown in Figure 1. First, the THPE was dissolved in water with a NaOH base, and dimethylmalonyl chloride was dissolved in the CHCl_3_. The reaction needed vigorous stirring to increase the surface area for the reaction, which enabled the formation of fine polyester nanoparticles. The polyester condensation reaction occured via the interaction between the phenoxide salt of the tris-phenoxide and the carbonyl carbon on the carbonyl chloride of dimethylmalonyl chloride at the interface of the chloroform–aqueous layers. The yield percentage of the reaction was 82%, The polyester swelling values were calculated to be 65.20% after 2 h and 160.70% after 24 h. This result shows that the polyester was hydrophilic.

### 2.2. Molecular Weight Measurement

The polyester was found to be insoluble in all common organic solvents. Therefore, an estimation of the molecular weight of the prepared insoluble polyester was measured in an aqueous suspension using dynamic light scattering (DLS). The fundamental quantity measured in a DLS-based instrument is particle diffusion. This diffusion coefficient (D) is related to hydrodynamic size via the Stokes–Einstein equation. The molecular weight was around 97,200 g/mol. The equation used to calculate the MW of the suspended polyester particle is as follows:MW = (α/D)^1/β^

The α constant is related to the specific composition of the diffusing particles and the surrounding solvent. The β depends on the temperature, solvent viscosity, and mass density of the particles.

### 2.3. FTIR Spectra Analysis

The FTIR spectrum for the CHPE nanoparticles showed the typical sharp band at 1726 cm^−1^ due to the C=O of the ester. The O-H end group was shown as a broad band from 3149 to 3603 cm^−1^. The aromatic CH showed a band at 2980 cm^−1^ and the aliphatic CH appeared as a band at 2952 cm^−1^, as shown in Figure 2a [32]. On the other hand, the FTIR spectrum of polyester–Cd (II) showed a sharp band at 1751 cm^−1^ due to the ester carbonyl group. The OH end group appeared as a broad band from 3123 to 3712 cm^−1^, and the aromatic CH showed a band at 2985 cm^−1^. The aliphatic CH occurred at 2942 cm^−1^, as shown in Figure 2b. When comparing the IR spectra of the polyester and polyester–Cd (II), the characteristic ester band at 1726 cm^−1^ was shifted to 1751 cm^−1^ because of the polyester bonding with the Cd(II) ions [33].

### 2.4. Solid NMR Spectrum Analysis

The polyester structure was also confirmed by solid ^13^C-NMR, as shown in Figure 3. The chemical shift of the formed ester carbonyl carbon appeared at around 172.4 ppm. The CH carbons of the aromatic rings in the THPE unit were observed at 121.2 and 109.8 ppm. The quaternary carbons of the aromatic rings in the THPE showed at 152.3 and 130.8 ppm. The quaternary carbon between the three aromatic rings was observed at 33.4 and the CH_3_ was observed at 22.3 ppm. The quaternary carbon of the malonyl unit appeared at 28.9 ppm and the CH_3_ appeared at 55.4 ppm [34,35].

### 2.5. Thermal Stability

The synthesized polyester had a high thermal stability. The TGA thermogram revealed a typical two-step breakdown profile, as shown in Figure 4. The thermogram showed one slight loss of mass and another significant loss. The first loss (only 9.8%) occurred between 180 and 200 °C, which could be explained by the decarboxylation of the carboxylic acid end groups. The second loss (65.0%) between 300 and 400 °C was due to the thermal degradation of the organic part of the polyester. 

### 2.6. XRD Analysis

The XRD analysis of the hyperbranched polyester nanoparticles provided important solid-state structural information, such as the degree of crystallinity, which can indicate the XRD pattern appearance. However, the XRD of the prepared polyester showed a semi-crystalline pattern, as evident by the presence of the peaks at positions ~18° and 20° (Figure 5) [36].

### 2.7. Particle Size Analysis

#### 2.7.1. Zeta Potential and Particle Size Analysis

The method used to synthesize the polyester and create nanoparticles was the interfacial polymerization common method. The formed polyester possessed a particles size of about 900 nm, as shown in Figure 6b. In addition, the polyester was characterized as a fine powder with a negative charge of around 21.41 mV, as shown in Figure 6a. These properties of the prepared polyester may increase its metal uptake from water due to the increased surface area of polyester resulting from the small particle size. In addition, the negative charge of the polyester surface increased the attractive force between Cd(II) cations and the polyester surface.

#### 2.7.2. SEM Analysis

SEM images of the polyester, as represented in Figure 7, showed irregular, dense microscale nano-aggregates. In Figure 7a, it was clear that the particles were within submicron size (below 1 µm), which is in line with the results obtained from the zeta sizer. In addition, small pores with an average size of <5 μm are seen on the surface. When the polyester was employed to adsorb Cd(II) ions from water, these pores provided suitable diffusion pathways for the Cd(II) ions to enter the polyester [9].

#### 2.7.3. TEM Analysis

TEM analysis was performed to obtain the shape and size distribution of the nanoparticles. Figure 8 shows the TEM images of the hyperbranched polyester nanoparticles. In TEM, the nanoparticles were observed to have a roughly spherical shape with a particle size range of 180–643 nm (Figure 8).

The particle size was determined using three different methods. In the DLS, the CHPE particle size was around 900 nm, while in TEM analysis, the CHPE particle size was between 180 nm and 643 nm. The variance between the particle sizes observed by these two methods was because the CHPE particles were swelled with water during DLS, as shown in Figure 9. On the other hand, the SEM images showed dense nano-aggregates and particles that were within submicron size (below 1 µm).

Both TEM and DLS provide complimentary, non-competing information about particle size. DLS is an intensity-based measurement that gives the hydrodynamic size, which is the size of the nanoparticle plus the liquid layer around the particle. In contrast, the size measured by TEM is the actual size of the nanoparticle. Also, note that electron microscopy examines the electron-dense part of the nanoparticles, while in contrast, DLS measures how the particle moves in suspension. Therefore, the hydrodynamic size reflects the particle’s transport properties and considers any protecting layers (surfactant, steric layer) or stabilizing layers that may surround the particle [37,38,39,40].

### 2.8. Metal Uptake Behavior of Cd(II) Ions from Water

#### 2.8.1. Kinetics Study of Cd(II) Ion Uptake by the Polyester

Polyester Cd(II) uptake was studied at different times and temperatures to understand the kinetic behavior of Cd(II) adsorption. A high sorption rate was observed during the initial 30 min; however, Cd(II) ion adsorption increased over time, reaching total saturation after 20 h of contact, as shown in Appendix A. Therefore, the polyester adsorption kinetics of Cd(II) were determined for an exposure time from 30 min to 48 h at 25, 35, and 45.0 °C with an initial concentration of 150 ppm for each metal ion. The results of these experiments are presented in plots in Figure 10a. 

The experiments fitted the pseudo-second-order model. The calculated amount of Cd(II) adsorbed at equilibrium (q_e_) values agreed very well with the experimental data (Table 1). This model was more suitable for representing kinetic data. This trend indicates that the rate-limiting stage for Cd (II) ions is chemisorption, which involves valence forces via electron sharing or exchange between the polyester and Cd(II) ions (Figure 10b). 

The amount of Cd(II) ion uptake by the CHBE nanoparticles (q_e_) was calculated by the following equation: $$\mathrm{qe}=\frac{\left(\mathrm{Initial}\;\mathrm{Cd}\left(\mathrm{II}\right)\;\mathrm{concentration}\;\left(\mathrm{Co}\right)-\mathrm{Residual}\;\mathrm{concentration}\;\mathrm{of}\;\mathrm{the}\;\mathrm{Cd}\left(\mathrm{II}\right)\left(\mathrm{Ceq}\right)\right)\mathrm{volume}\;\mathrm{of}\;\mathrm{solution}}{\mathrm{Mass}\;\mathrm{of}\;\mathrm{CHBE}\;\mathrm{nanoparticles}\;}$$

The following equation was used to calculate the loading percentage of Cd(II) on the CHBE nanoparticles: $$\mathrm{Cd}\left(\mathrm{II}\right)\;\mathrm{uptake}\;\left(\%\right)=\frac{\;\mathrm{Residual}\;\mathrm{concentration}\;\mathrm{of}\;\mathrm{the}\;\mathrm{Cd}\left(\mathrm{II}\right)\left(\mathrm{Ceq}\right)}{\mathrm{Initial}\;\mathrm{Cd}\left(\mathrm{II}\right)\;\mathrm{concentration}\;\left(\mathrm{Co}\right)}$$

The activation energy (E_a_) was calculated using the Arrhenius equation as follows: ln k = −E_a_ /R T + ln A
where k is the rate constant; E_a_ is the activation energy; R is the gas constant; and A is the pre-exponential factor.

The value of E_a_ was measured from the intercept and slope of a straight line of a plot of ln k against 1/T. The E_a_ value for Cd(II) was 63.11 kJ/mol. E_a_ was approximately constant over a moderate temperature of 25 K (Figure 10c). 

Hyperbranched polyester nanoparticles had better Cd(II) adsorption capacities than a linear polyester with chelating groups. This may be due to the three-dimensional structure [10]. In addition, it also showed higher adsorption capacity that an aliphatic hyperbranched polyester, which is because of the nanoparticle size and the malonate chelating group in the polymer chain [37] (Table 2). 

#### 2.8.2. Thermodynamics of Adsorption

The effect of temperature on the uptake percentage of Cd(II) ion adsorption was also studied. The adsorption process of Cd(II) onto the surface of the polyester was endothermic, since the uptake increased as the temperature increased at the same pH value, as shown in Figure 11a. The thermodynamic functions ΔG, ΔH, and ΔS were related to the experimental conditions. 

The distribution coefficient (Kd) was calculated using the following equation:


$$Kd=\frac{\text{ metal ion on the polymer }(mg)\times \text{ volume of the solution }(L)}{\text{ metal ion in solution }(mg)\times \text{ mass of polymer }(g)}$$


The values of ΔH° and ΔS° were calculated from the slopes and intercepts of the plot of ln Kd vs. 1/T using the following equation: ln Kd = (ΔS°/R) − (ΔH°/RT)

The change in Gibbs free energy (ΔG°) was calculated using the following equation:ΔG° = ΔH − TΔS

The value of ΔG was −0.91 kJ/mol, indicating the adsorption process’s favorability. Conversely, a lower value of ΔG would suggest that the process of adsorption was more favorable at high temperatures. The positive ΔH (7.22 kJ/mol) value indicated that the metal adsorption was endothermic. The value of ΔS was 27.27 J/mol·K; the positive value showed the increased randomness at the solid–solution interface during the adsorption process due to dehydration, since the removal of water from ions is essentially an endothermic process. The positive entropy of adsorption also reflected the affinity of the adsorbent for the metal ions used. 

#### 2.8.3. Desorption Studies

Two eluting agents—the proton-exchanging agent 0.1 M HNO_3_ and the complex-forming agent 0.1 M EDTA—were used to remove Cd(II) ions and regenerate the polyester, keeping the flow rate of elution at 1 mL/min. The eluate was collected in five fractions of 10 mL each; the results are expressed as percent recovery and represented in Table 3. The cumulative recovery present for Cd(II) is slightly higher when using 0.1 M HNO_3_, which supports the idea that the sorption is physical in nature and the desorption is through cation exchange. 

## 3. Experimental Section

### 3.1. Materials

The dimethylmalonyl chloride, sodium perchlorate, 1,1,1-tris(4-hydroxyphenyl)ethane (THPE), and 37% hydrochloric acid were obtained from Acros; the sodium hydroxide was obtained from LOBA; the pyridine was obtained from GPR; the methanol was obtained from Fluka; and the anhydrous chloroform and cadmium nitrate tetrahydrate were obtained from BDH. 

### 3.2. Synthesis of Hyperbranched Polyester Nanoparticles

The polyester was synthesized from its monomers using interfacial polymerization in chloroform and water as follows: THPE (1.23 g, 4 mmol) was dissolved into water (15 mL) in a 150 mL round-bottom flask. NaOH (0.48 g, 12 mmol) equivalent to tris-phenol was added to convert the phenolic group to sodium phenoxide salt. Dimethylmalonyl chloride (1.02 g, 6 mmol) was dissolved in 10 mL of dry chloroform and added dropwise to the phenoxide salt over 15 min. The reaction mixture was stirred at 0–7 °C for 15 min and then for 1 h at room temperature, where the formed polyester was precipitated. The resulting polyester was filtered and washed with methanol. The white polyester powder was dried by a vacuum oven at 70 °C for five hours (82% yield).

### 3.3. Swelling Capacity 

An amount of 1.00 g of dry polyester was placed in 100 mL of distilled water. The swelled polyester was filtered and weighed after 2 and 24 h. The value of the swelling capacity was calculated using the following equation:$$\mathrm{Swelling}\;\mathrm{capacity}\;=\frac{\mathrm{Polyester}\;\mathrm{mass}\;\mathrm{after}\;\mathrm{filtration}\;-\mathrm{Polyester}\;\mathrm{mass}\;\mathrm{after}\;\mathrm{drying}}{\mathrm{Polyester}\;\mathrm{mass}\;\mathrm{after}\;\mathrm{filtration}\;}$$

### 3.4. Characterization 

#### 3.4.1. Particle Size, Zeta Potential Analysis, and Molecular Weight Measurement 

The polyester’s average diameter, zeta potential, and molecular weight were analyzed at 25 °C in an aqueous suspension by dynamic light scattering using a Nicomp N3000 (Billerica, MA, USA).

#### 3.4.2. Fourier Transform Infrared (FTIR) Spectroscopy Analysis 

The polyester and polyester–Cd(II) FTIR spectra were recorded using a Perkin Elmer FTIR spectrometer (Akron, OH, USA), coupled with the Spectrum 10 software to operate and treat the FTIR spectra.

#### 3.4.3. Solid Nuclear Magnetic Resonance (NMR) Spectroscopy 

The prepared polyester solid ^13^C-NMR spectrum was recorded on a Bruker 500 MHz spectrometer (Ettlingen, Germany).

#### 3.4.4. Thermal Gravimetric Analysis (TGA)

The TGA of the polyester was performed using a Netzsch STA 409 PG/PC thermal analyzer (Selb Bavaria, Germany). A heating rate of 20 °C/min was used in a dry nitrogen environment purged at a 50 mL/min flow rate.

#### 3.4.5. X-ray Diffraction (XRD) Analysis

X-ray diffractometry (MiniFlex 600 benchtop diffractometer (RigaKu, Tokyo, Japan)) was used to investigate the physical form of the polyester. The XRD experiments were performed over the range 2θ from 5 to 99°, with Cu Kα radiation (1.5148227 Å). Data were recorded at a scanning speed of 5 °/minute. The OriginPro® software was used to analyze the scans (Northampton, MA, USA).

#### 3.4.6. Scanning Electron Micrographs (SEM)

Surface topography was examined for the polyester using JSM-IT300 (JEOL, Tokyo, Japan). An amount of 2–4 mg of each sample was sprinkled on a double-adhesive carbon tape affixed to an aluminum tub. Samples were analyzed without any further coating.

#### 3.4.7. Transmission Electron Microscopy (TEM)

One drop of fresh NP dispersion was placed on a Formvar-coated copper grid (300 mesh, Electron Microscopy Sciences, Hatfield, PA, USA). Excess liquid was wiped with the tip of a filter paper after 1 min, and the grid was allowed to dry overnight. At a 60 kV accelerating voltage, imaging was accomplished using an FEI Morgagni 268 TEM (Eindhoven, The, Netherlands).

### 3.5. The Removal of Cd(II) Ions from Water

#### 3.5.1. Preparation of Stock Solutions of Metal

Stock solutions of Cd(II) ions (1000.0 mg/L) were made by dissolving appropriate quantities of Cd(II) salt in 0.1 M NaClO_4_, which were then adjusted to a pH of 4.00 using a METROHM 605 pH meter. Stock solutions were used to make solutions with various concentrations (150.0, 125.0, 100.0, 50.0, 25.0, and 15.0 mg/L). The dilutions were made with 0.10 M NaClO_4_ and adjusted with 0.10 M HClO_4_ to reach a target pH of 4.00.

#### 3.5.2. Metal Uptake Characteristics of the Polymer by Batch Experiments

The metal uptake characteristics for Cd(II) ions were studied using the batch equilibrium technique. An aqueous solution of known Cd(II) concentration (25.0 mL) was added to the polymer powder (0.100 g), and the mixed solutions were mechanically shaken using a Clifton Shaker (BS-11) equipped with a thermostat. After a certain period at a temperature of 25 °C, the mixture was filtered. The amount of Cd(II) ions remaining in the filtrate solution was determined using a Varian Spectra AA-250 pulse atomic absorption spectrometer after constructing an analytical calibration curve for the metal ions (Cd(II)).

#### 3.5.3. Desorption Studies

The Cd(II) ions were desorbed using a column experiment loaded with Cd (II), then eluted with 0.1 M HNO_3_ or 0.1 M EDTA at an elution flow rate of 0.2 mL/min. A Varian Spectra AA-250 pulse atomic absorption spectrometer was used to quantify the concentration of Cd(II) in the eluate collected in five distinct portions (10.0 mL each). Every measurement was done three times to ensure that the results were accurate and reproducible, and then the average was calculated using the following equation:$$\%\mathrm{Recovery}=\frac{\mathrm{Amount}\;\mathrm{of}\;\mathrm{eluted}\;\left(\mathrm{mg}\right)}{\mathrm{Total}\;\mathrm{amount}\;\mathrm{adsorbed}\;\mathrm{by}\;\mathrm{the}\;\mathrm{polymer}\;\left(\mathrm{mg}\right)}\times 100\%.$$

## 4. Conclusions

In the present study, hyperbranched polyester nanoparticles were successfully synthesized using the interfacial polycondensation method. The particles were found to be highly efficient during the removal of cadmium metal ions, since the adsorption capacity was as high as 27 mg/g at 25 °C, 37 mg/g at 35 °C, and 38.9 mg/g at 45 °C as result of the good hydrophilic properties, negative surface, and nanosized particles. In addition, it is easy to regenerate using dilute acid (0.10 M). The prepared polyester is an effective and efficient Cd(II) adsorbent for water purification applications.

## Figures and Tables

**Figure 1 molecules-27-03656-f001:**
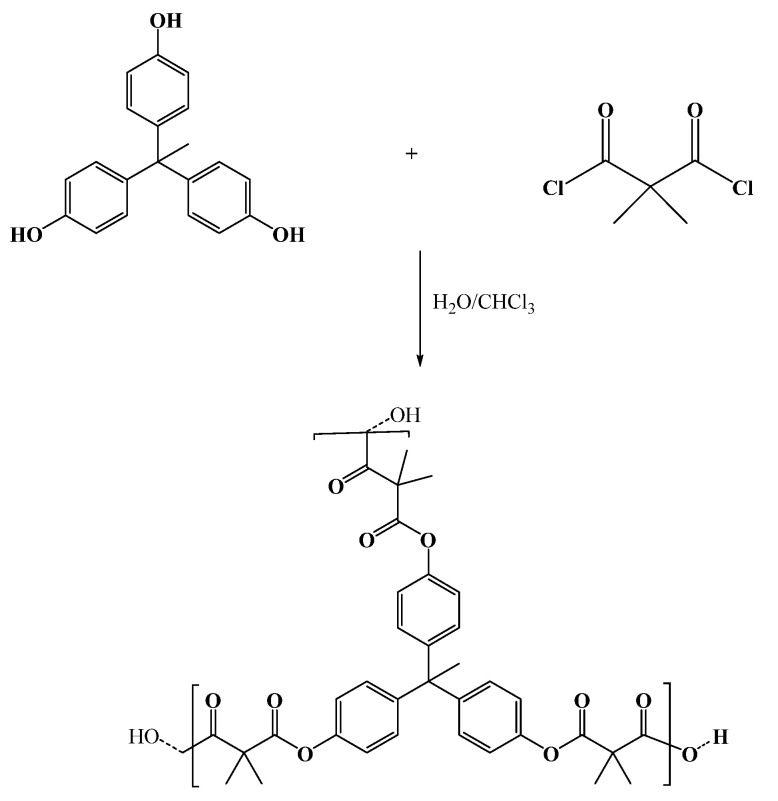
Synthesis of a hyperbranched polyester based on THPE.

**Figure 2 molecules-27-03656-f002:**
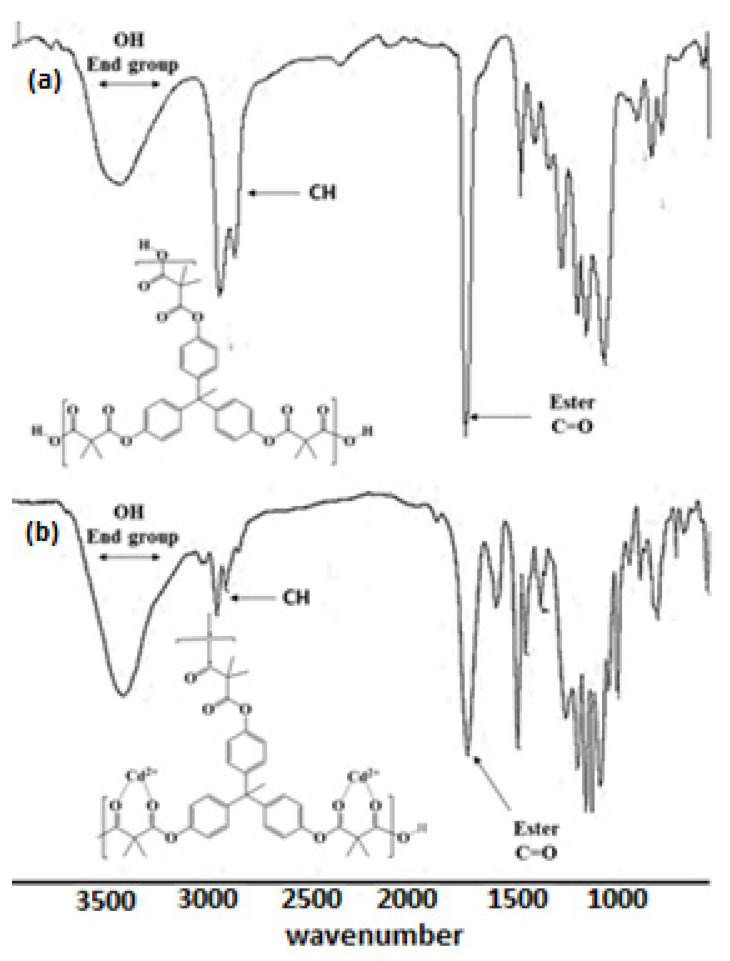
FTIR spectra of (**a**) the hyperbranched polyester and (**b**) the hyperbranched polyester–Cd (II).

**Figure 3 molecules-27-03656-f003:**
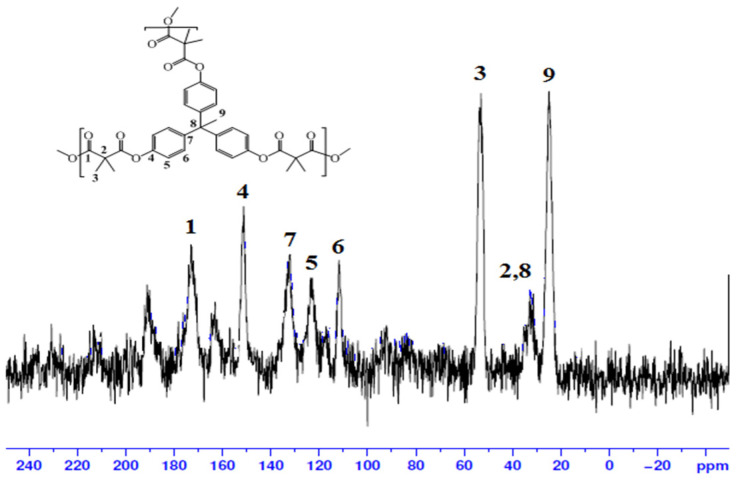
Solid NMR spectrum of the hyperbranched polyester.

**Figure 4 molecules-27-03656-f004:**
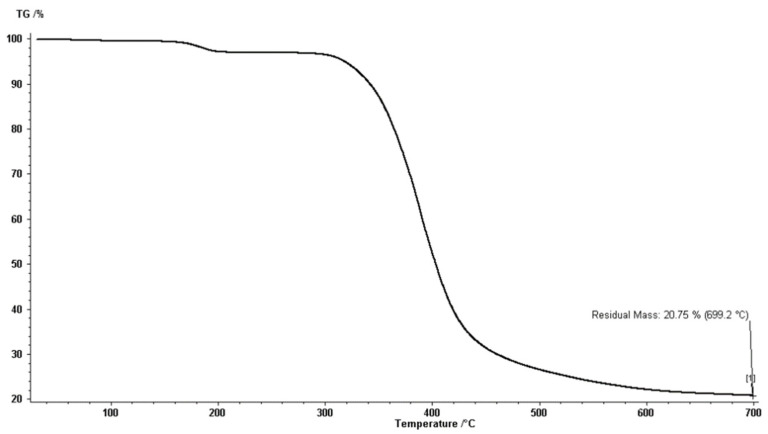
TGA thermogram of the hyperbranched polyester.

**Figure 5 molecules-27-03656-f005:**
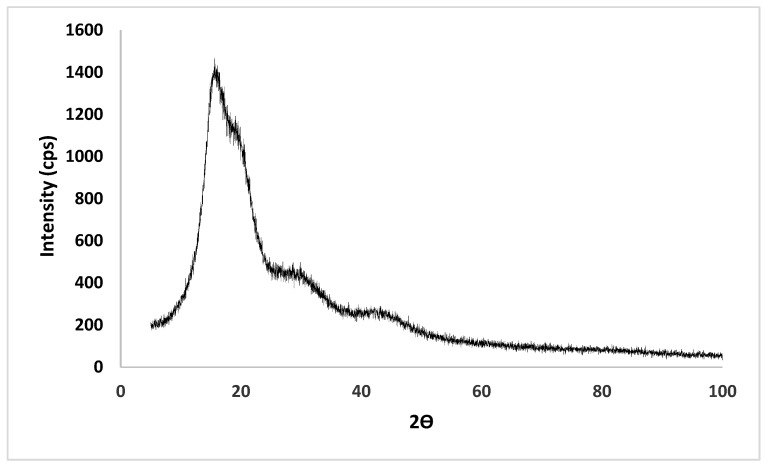
X-ray diffraction patterns for the hyperbranched polyester powder.

**Figure 6 molecules-27-03656-f006:**
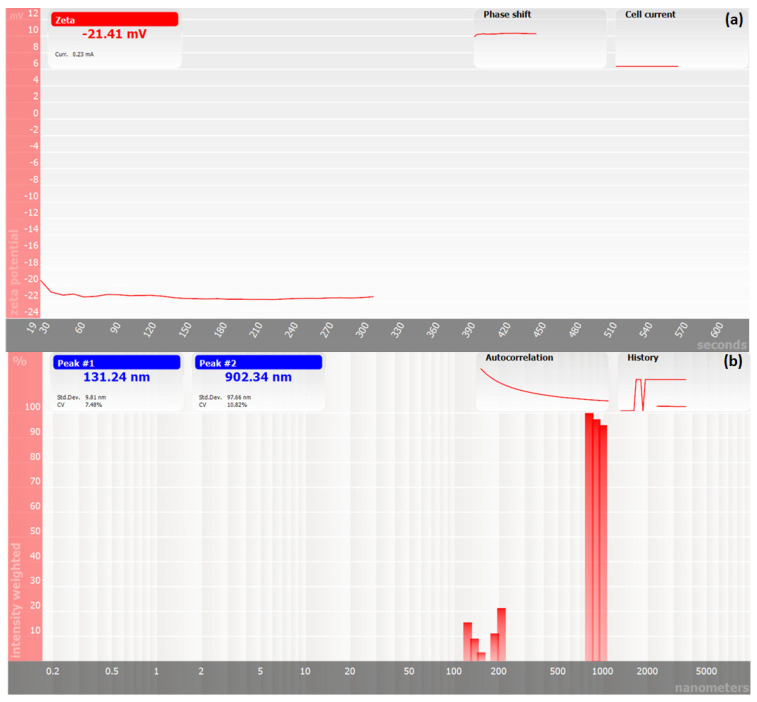
(**a**) The zeta potential (mV) and (**b**) the particle size (nm).

**Figure 7 molecules-27-03656-f007:**
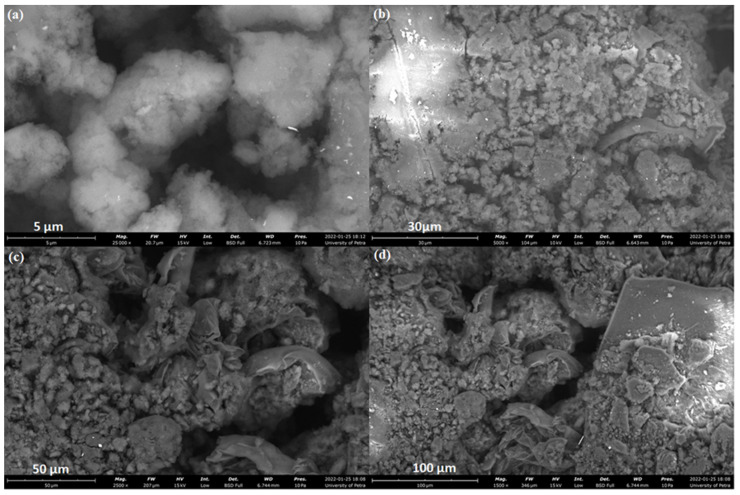
SEM micrographs of the hyperbranched polyester (**a**) 5 µm (**b**) 30 µm (**c**) 50 µm (**d**) 100 µm.

**Figure 8 molecules-27-03656-f008:**
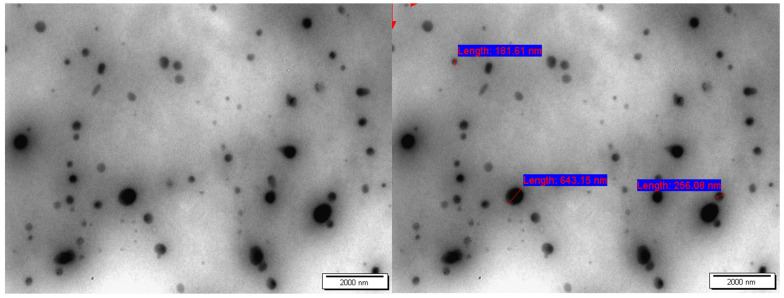
TEM images of the hyperbranched polyester.

**Figure 9 molecules-27-03656-f009:**
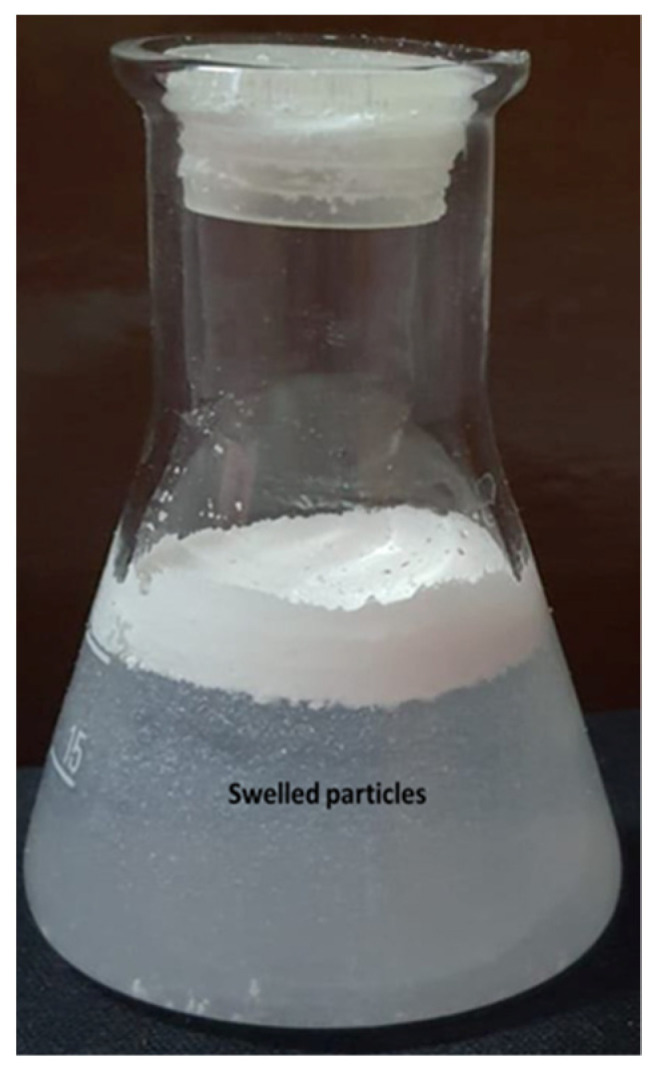
Dispersion of the CHPE polyester immediately after adding 0.5 g to 25 mL of water.

**Figure 10 molecules-27-03656-f010:**
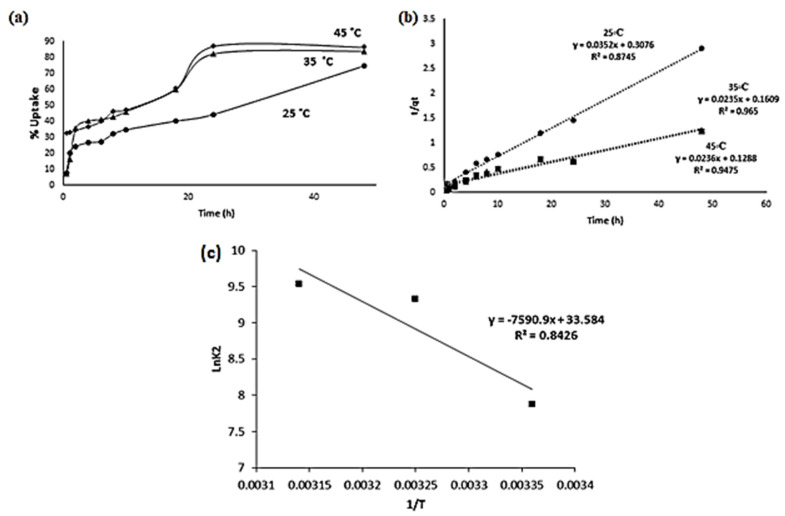
(**a**) Cd(II) uptake as a function of contact time, at pH = 4.00; (**b**) the pseudo-second-order sorption kinetics of Cd(II) as a function of contact time at pH = 4.00; and (**c**) the Arrhenius plots of Cd(II) at pH = 4.00.

**Figure 11 molecules-27-03656-f011:**
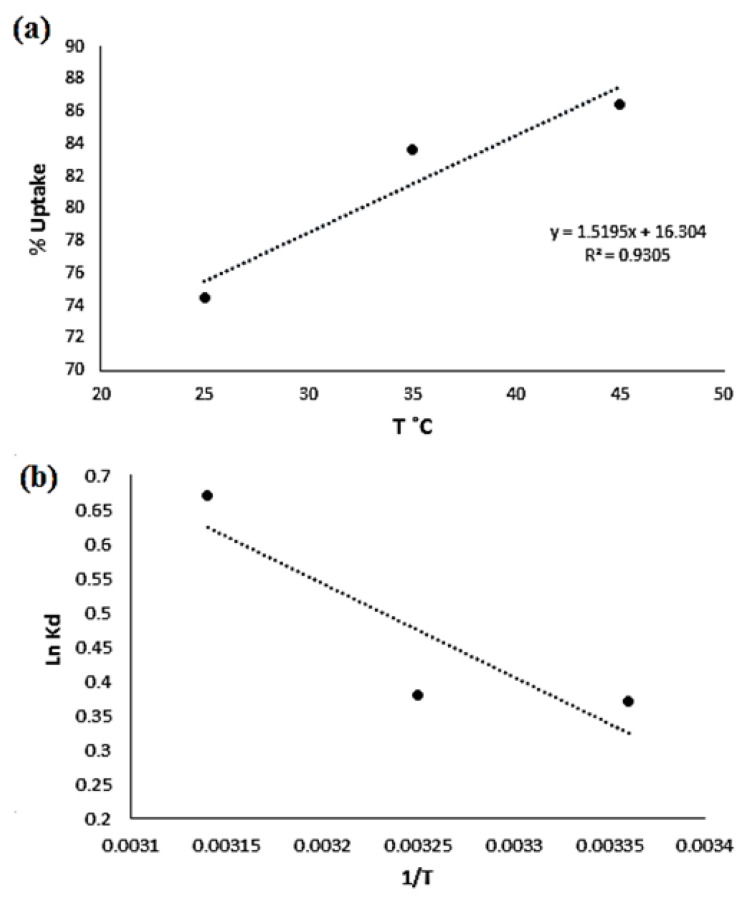
(**a**) The % uptake of Cd(II) vs. temperature; (**b**) a plot of lnK_d_ vs. 1/T(K) for Cd(II) and Fe(III) at pH = 4.00.

**Table 1 molecules-27-03656-t001:** The pseudo-second-order reaction kinetics for the uptake of Cd(II) ions by polyester at 25 °C, 35 °C, and 45 °C.

T (°C)	Adsorption Rate Constants (k)	Calculated Adsorbed at Equilibrium (q_e_ cal)	Experimental Adsorbed at Equilibrium (q_e_ exp)
25	2.62 × 10^3^	28.41	27.93
35	1.13 × 10^4^	42.55	36.89
45	1.39 × 10^4^	42.37	39.13

**Table 2 molecules-27-03656-t002:** Cd(II) adsorption capacities of different adsorbents.

Adsorbent	Adsorption Capacities (mg/g)	Reference
Linear polyester	3.0	[10]
Aliphatic Hyperbranched polyester	0.29	[37]
Hyperbranched polyester nanoparticles	27	This work

**Table 3 molecules-27-03656-t003:** Desorption of Cd(II) of polyester.

Eluting Agent	% Recovery					% Cumulative Recovery
	First Fraction	Second Fraction	Third Fraction	Fourth Fraction	Fifth Fraction	
0.1 M HNO_3_	10.50	5.87	3.37	2.34	1.80	23.88
0.1 M EDTA	4.32	4.02	4.02	4.02	3.92	20.30

## Data Availability

All data related to this work are presented in the manuscript.

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
