# Peer review of "Synthesis and Characterization of Chelating Hyperbranched Polyester Nanoparticles for Cd(II) Ion Removal from Water"

_molecules, 2022, doi:10.3390/molecules27123656_

Round 1
Reviewer 1 Report
I'm content in current form.
Author Response
Thank you for your comments which have improved the clarity of the manuscript.
Reviewer 2 Report
The manuscript entitled "Synthesis and characterization of chelating hyperbranched polyester nanoparticles and studying its metal uptake behavior towards Cd(II) ions from water" aims to present the synthesis and characterization of chelating hyperbranched polyester nanoparticles prepared by interfacial polycondensation, and use it to remove metal ions from water. The manuscript also studies nanoparticles' uptake potential towards Cd(II) ions as a function of temperature, contact time, and concentration. The process's thermodynamics and kinetics, and desorption data on the polyester were also studied. The manuscript is greatly improved compared to its first version, but some improvements are still to be made.
I suggest the title as follows: "Synthesis and characterization of chelating hyperbranched polyester nanoparticles for Cd(II) ions removal from water. "
Table 1 should be transferred to Supplementary material.
Author Response
- The title was changed to Synthesis and characterization of chelating hyperbranched polyester nanoparticles for Cd(II) ions removal from water
- Table 1 was transferred to Supplementary material.
Reviewer 3 Report
The Authors revised the manuscript well. Now it is acceptable for publicatrion as is.
Author Response

(The authors gave the same response as above.)
